# RECAP: Towards Precise Radiology Report Generation via Dynamic Disease Progression Reasoning

**Wenjun Hou**[1,2], **Yi Cheng**[1*], **Kaishuai Xu**[1*], **Wenjie Li**[1†], **Jiang Liu**[2†]

[1]Department of Computing, The Hong Kong Polytechnic University, HKSAR, China

[2]Research Institute of Trustworthy Autonomous Systems and
Department of Computer Science and Engineering,
Southern University of Science and Technology, Shenzhen, China

houwenjun060@gmail.com
{alyssa.cheng, kaishuaii.xu}@connect.polyu.hk
cswjli@comp.polyu.edu.hk, liuj@sustech.edu.cn

## Abstract

Automating radiology report generation can significantly alleviate radiologists' workloads. Previous research has primarily focused on realizing highly concise observations while neglecting the precise attributes that determine the severity of diseases (e.g., _small pleural effusion_). Since incorrect attributes will lead to imprecise radiology reports, strengthening the generation process with precise attribute modeling becomes necessary. Additionally, the temporal information contained in the historical records, which is crucial in evaluating a patient's current condition (e.g., _heart size is unchanged_), has also been largely disregarded. To address these issues, we propose RECAP, which generates precise and accurate radiology reports via dynamic disease progression reasoning. Specifically, RECAP first predicts the observations and progressions (i.e., spatiotemporal information) given two consecutive radiographs. It then combines the historical records, spatiotemporal information, and radiographs for report generation, where a disease progression graph and dynamic progression reasoning mechanism are devised to accurately select the attributes of each observation and progression. Extensive experiments on two publicly available datasets demonstrate the effectiveness of our model.[1]

## 1 Introduction

Radiology report generation (Rennie et al., 2017; Anderson et al., 2018; Chen et al., 2020), aiming to generate clinically coherent and factually accurate free-text reports, has received increasing attention

---

[*]Equal Contribution.

[†]Corresponding authors.

[1]Our code is available at https://github.com/wjhou/Recap.

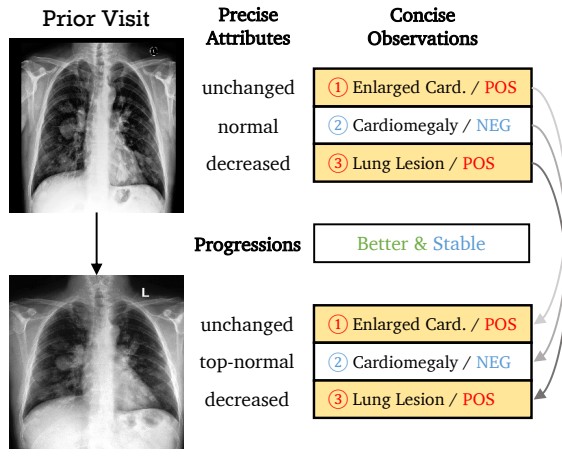

Figure 1: An example of a follow-up visit record with its prior visit record. Part of their observations are listed with their precise attributes. _Enlarged Card._ denotes _Enlarged Cardiomediastinum_.

from the research community due to its large potential to alleviate radiologists' workloads.

Recent research works (Nooralahzadeh et al., 2021; Nishino et al., 2022; Delbrouck et al., 2022; Bannur et al., 2023; Tanida et al., 2023; Hou et al., 2023) have made significant efforts in improving the clinical factuality of generated reports. Despite their progress, these methods still struggle to produce precise and accurate free-text reports. One significant problem within these methods is that although they successfully captured the semantic information of observations, their attributes still remain imprecise. They either ignored historical records (i.e., temporal information) that are required for assessing patients' current conditions or omitted the fine-grained attributes of observations (i.e., spatial information) that are crucial in quantifying the severity of diseases, which are far from adequate and often lead to imprecise reports. Both

temporal and spatial information are crucial for generating precise and accurate reports. For instance, as illustrated in Figure 1, the patient's conditions can change from time to time, and the observations become `Better` and `Stable`. Only if accessing the historical records, the overall conditions could be estimated. In addition, different attributes reflect the severity of an observation, such as *normal* and *top-normal* for *Cardiomegaly*. In order to produce precise and accurate free-text reports, we must consider both kinds of information and apply stronger reasoning to strengthen the generation process with precise attribute modeling.

In this paper, we propose RECAP, which captures both temporal and spatial information for radiology REport Generation via DynamiC DiseAse Progression Reasoning. Specifically, RECAP first predicts observations and progressions given two consecutive radiographs. It then combines them with the historical records and the current radiograph for report generation. To achieve precise attribute modeling, we construct a disease progression graph, which contains the prior and current observations, the progressions, and the precise attributes. We then devise a dynamic progression reasoning (PrR) mechanism that aggregates information in the graph to select observation-relevant attributes.

In conclusion, our contributions can be summarized as follows:

- We propose RECAP, which can capture both spatial and temporal information for generating precise and accurate free-text reports.

- To achieve precise attribute modeling, we construct a disease progression graph containing both observations and fine-grained attributes that quantify the severity of diseases. Then, we devise a dynamic disease progression reasoning (PrR) mechanism to select observation/progression-relevant attributes.

- We conduct extensive experiments on two publicly available benchmarks, and experimental results demonstrate the effectiveness of our model in generating precise and accurate radiology reports.

## 2 Preliminary

### 2.1 Problem Formulation

Given a radiograph-report pair $D^c = \{X^c, Y^c\}$, with its record of last visit being either $D^p =$

$\{X^p, Y^p\}$ or $D^p = \emptyset$ if the historical record is missing[2], the task of radiology report generation aims to maximize $p(Y^c|X^c, D^p)$. To learn the spatiotemporal information, observations $O$ (i.e., spatial information) (Irvin et al., 2019) and progressions $P$ (i.e., temporal information) (Wu et al., 2021) are introduced. Then, the report generation process is divided into two stages in our framework, i.e., observation and progression prediction (i.e., Stage 1) and spatiotemporal-aware report generation (i.e., Stage 2). Specifically, the probability of observations and progressions are denoted as $p(O|X^c)$ and $p(P|X^c, X^p)$, respectively, and then the generation process is modeled as $p(Y^c|X^c, D^p, O, P)$. Finally, our framework aims to maximize the following probability:

$$p(Y^c|X^c, D^p) \propto \overbrace{p(O|X^c) \cdot p(P|X^c, X^p)}^{\text{Stage 1}} \cdot \underbrace{p(Y^c|X^c, D^p, O, P)}_{\text{Stage 2}}.$$

### 2.2 Progression Graph Construction

**Observation and Progression Extraction.** For each report, we first label its observations $O = \{o_1, \ldots, o_{|o|}\}$ with CheXbert (Smit et al., 2020). Similar to Hou et al. (2023), each observation is further labeled with its status (i.e., *Positive*, *Negative*, *Uncertain*, and *Blank*). We convert *Positive* and *Uncertain* as POS, *Negative* as NEG, and remove *Blank*, as shown in Figure 1. Then, we extract progression information $P$ of a patient with Chest ImaGenome (Wu et al., 2021) which provides progression labels (i.e., `Better`, `Stable`, or `Worse`) between two regions of interest (ROIs) in $X^p$ and $X^c$, respectively. However, extracting ROIs could be difficult, and adopting such ROI-level labels may not generalize well across different datasets. Thus, we use image-level labels, which only indicate whether there are any progressions between $X^p$ and $X^c$. As a result, a patient may have different progressions (e.g., both `Better` and `Worse`). The statistics of observations and progressions can be found in Appendix A.1.

**Spatial/Temporal Entity (Attribute) Collection.**[3] To model spatial and temporal information, we

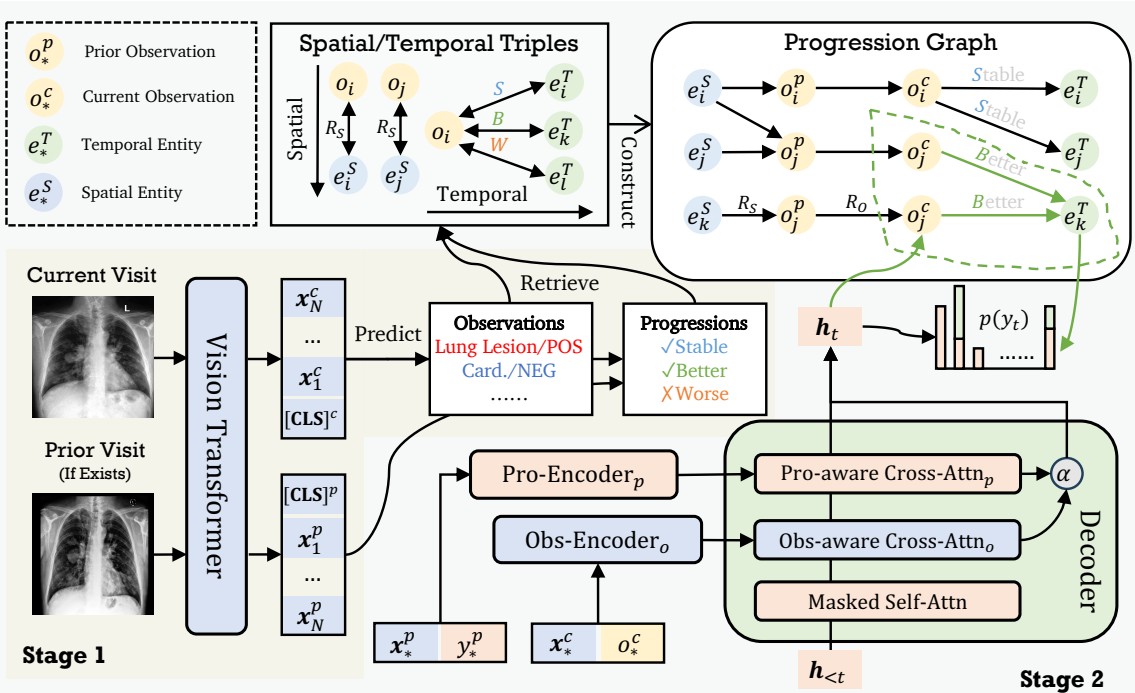

---
[2]There are two kinds of records (i.e., first-visit and follow-up-visit). If it is the first visit of a patient, the historical record does not exist.

[3]Attributes are included in the entity set as provided by Jain et al. (2021). For simplicity, we use "attribute" and "entity" interchangeably in this paper.

Figure 2: Overview of the RECAP framework. *Pro-Encoder$_p$* is the progression-related encoder and *Obs-Encoder$_o$* is the observation-related encoder, respectively. Other modules in the decoder are omitted for simplicity.

collect a set of entities to represent it. For temporal entities, we adopt the entities provided by (Bannur et al., 2023), denoted as $E^T$. For spatial entities $E^S$, we adopt the entities with a relation *modify* or *located_at* in RadGraph (Jain et al., 2021), and we also filter out stopwords[4] and temporal entities from them. Part of the temporal and spatial entities are listed in Appendix A.2.

**Progression Graph Construction.** Our progression graph $G = <V, R>$ is constructed based purely on the training corpus in an unsupervised manner. Specifically, $V = \{O, E^T, E^S\}$ is the node-set, and $R = \{S, B, W, R_S, R_O\}$ is the edge set, where $S$, $B$, and $W$ denote three progressions Stable, Better, and Worse, connecting an observation with an temporal entity. In addition, $R_s$ and $R_o$ are additional relations connecting current observations with spatial entities and prior/current observations, respectively. To extract spatial/temporal triples automatically, we use the proven-efficient statistical tool, i.e., pointwise mutual information (PMI; Church and Hanks (1990)), where a higher PMI score implies two units with higher co-occurrence, similar to Hou et al. (2023):

$$\text{PMI}(\bar{x}, \hat{x}) = \log \frac{p(\bar{x}, \hat{x})}{p(\bar{x})p(\hat{x})} = \log \frac{p(\hat{x}|\bar{x})}{p(\hat{x})},$$

Specifically, we set $\bar{x}$ to $(o_i, r_j)$ where $r_j \in R$ and

set $\hat{x}$ to $e_k^*$ where $e_k^* \in \{E^T, E^S\}$. Then, we rank these triples using $\text{PMI}((o_i, r_j), e_k^*)$ and select top-$K$ of them as candidates for each $(o_i, r_j)$. Finally, we use observations as the query to retrieve relevant triples. We consider edges in the graph: $e_i^* \xrightarrow{r_j} o_k^p \xrightarrow{R_O} o_l^c \xrightarrow{r_m} e_n^*$, as shown in the top-right of Figure 2, consistent with the progression direction.

## 3 Methodology

### 3.1 Visual Encoding

Given an image $X^c$, an image processor is first to split it into $N$ patches, and then a visual encoder (i.e., ViT (Dosovitskiy et al., 2021)) is adopted to extract visual representations $\boldsymbol{X}^c$:

$$\boldsymbol{X}^c = \{[\textbf{CLS}]^c, \boldsymbol{x}_1^c, \dots, \boldsymbol{x}_N^c\} = \text{ViT}(X^c),$$

where $[\textbf{CLS}]^c \in \mathbb{R}^h$ is the representation of the class token [CLS] prepended in the patch sequence, $\boldsymbol{x}_i^c \in \mathbb{R}^h$ is the $i$-th visual representation. Similarly, the visual representation of image $X^p$ is extracted using the same ViT model and represented as $\boldsymbol{X}^p = \{[\textbf{CLS}]^p, \boldsymbol{x}_1^p, \dots, \boldsymbol{x}_N^p\}$.

### 3.2 Stage 1: Observation and Progression Prediction

**Observation Prediction.** As observations can be measured from a single image solely, we only use the pooler output $[\textbf{CLS}]^c$ of $X^c$ for observation

prediction. Inspired by Tanida et al. (2023), we divide it into two steps, i.e., detection and then classification. Specifically, the detection probability $p_d(o_i)$ of the $i$-th observation presented in a report and the probability of this observation $p_c(o_i)$ being classified as abnormal are modeled as:

$$p_d(o_i) = \sigma(\boldsymbol{W}_{d_i}[\mathbf{CLS}]^c + b_{d_i}),$$
$$p_c(o_i) = \sigma(\boldsymbol{W}_{c_i}[\mathbf{CLS}]^c + b_{c_i}),$$

where $\sigma$ is the Sigmoid function, $\boldsymbol{W}_{d_i}, \boldsymbol{W}_{c_i} \in \mathbb{R}^h$ are the weight matrices and $b_{d_i}, b_{c_i} \in \mathbb{R}$ are the biases. Finally, the probability of the $i$-th observation is denoted as $p(o_i) = p_d(o_i) \cdot p_c(o_i)$. Note that for observation *No Finding* $o_n$ is presented in every sample, i.e., $p_d(o_n) = 1$ and $p(o_n) = p_c(o_n)$.

**Progression Prediction.** Similar to observation prediction, the pooler outputs $[\mathbf{CLS}]^p$ of $X^p$ and $[\mathbf{CLS}]^c$ of $X^c$ are adopted for progression prediction, and the probability of the $j$-th progression $p(p_j)$ is modeled as:

$$[\mathbf{CLS}] = [[\mathbf{CLS}]^p; [\mathbf{CLS}]^c],$$
$$p(p_j) = \sigma(\boldsymbol{W}_j[\mathbf{CLS}] + b_j),$$

where $[;]$ is the concatenation operation, $\boldsymbol{W}_j \in \mathbb{R}^{2h}$ is the weight matrix, and $b_j \in \mathbb{R}$ are the bias. As we found that learning sparse signals from image-level progression labels is difficult and has side effects on the performance of observation prediction, we detach $[\mathbf{CLS}]$ from the computational graph while training.

**Training.** We optimize these two prediction tasks by minimizing the binary cross-entropy loss. Specifically, the loss of observation detection $\mathcal{L}_d$ is denoted as:

$$\mathcal{L}_d = -\frac{1}{|O|} \sum [\alpha_d \cdot l_{d_i} \cdot \log p_d(o_i)$$
$$+ (1 - l_{d_i}) \cdot \log(1 - p_d(o_i))],$$

where $\alpha_d$ is the weight to tackle the class imbalance issue, $l_{d_i}$ denotes the label of $i$-th observation $d_i$. Similarly, the loss of observation classification $\mathcal{L}_c$ and progression prediction $\mathcal{L}_p$ can be calculated using the above equation. Note that $\mathcal{L}_c$ and $\mathcal{L}_p$ are unweighted loss. Finally, the overall loss of Stage 1 is $\mathcal{L}_{S1} = \mathcal{L}_d + \mathcal{L}_c + \mathcal{L}_p$.

### 3.3 Stage 2: SpatioTemporal-aware Report Generation

**Observation-aware Visual Encoding.** To learn the observation-aware visual representations, we jointly encode $\boldsymbol{X}^c$ and its observations $O^c$ using a Transformer encoder (Vaswani et al., 2017). Additionally, a special token [FiV] for first-visit records

or [FoV] for follow-up-visit records is appended to distinguish them, represented as [F*V]:

$$\boldsymbol{h}^c = [\boldsymbol{h}_X^c; \boldsymbol{h}_o^c] = \text{Encoder}_o([\boldsymbol{X}^c; [\text{F*V}]; O^c]),$$

where $\boldsymbol{h}_X^c, \boldsymbol{h}_o^c \in \mathbb{R}^h$ are the visual hidden representations and observation hidden representations of the current radiograph and observations.

**Progression-aware Information Encoding.** We use another encoder to encode the progression information (i.e., temporal information). Specifically, given $\boldsymbol{X}^p$ and $Y^p$, the hidden states of the prior record are represented as:

$$\boldsymbol{h}^p = [\boldsymbol{h}_X^p; \boldsymbol{h}_Y^p] = \text{Encoder}_p([\boldsymbol{X}^p; Y^p]),$$

where $\boldsymbol{h}_X^p, \boldsymbol{h}_Y^p \in \mathbb{R}^h$ are the visual hidden representations and textual hidden representations of prior records, respectively.

**Concise Report Decoding.** Given $\boldsymbol{h}^p$ and $\boldsymbol{h}^c$, a Transformer decoder is adopted for report generation. Since not every sample has a prior record and follow-up records may include new observations, controlling the progression information is necessary. Thus, we include a soft gate $\alpha$ to fuse the observation-related and progression-related information, as shown in Figure 2:

$$\text{Decoder} = \begin{cases} \boldsymbol{h}_t^s = \text{Self-Attn}(\boldsymbol{h}_t^w, \boldsymbol{h}_{<t}^w, \boldsymbol{h}_{<t}^w), \\ \tilde{\boldsymbol{h}}_t^c = \text{Cross-Attn}_o(\boldsymbol{h}_t^s, \boldsymbol{h}^c, \boldsymbol{h}^c), \\ \tilde{\boldsymbol{h}}_t^p = \text{Cross-Attn}_p(\tilde{\boldsymbol{h}}_t^c, \boldsymbol{h}^p, \boldsymbol{h}^p), \\ \alpha = \sigma(\boldsymbol{W}_\alpha \tilde{\boldsymbol{h}}_t^c + b_\alpha), \\ \boldsymbol{h}_t = \alpha \cdot \tilde{\boldsymbol{h}}_t^p + (1 - \alpha) \cdot \tilde{\boldsymbol{h}}_t^c, \end{cases}$$
$$p_{\mathcal{V}}(y_t) = \text{Softmax}(\boldsymbol{W}_{\mathcal{V}} \boldsymbol{h}_t + \boldsymbol{b}_{\mathcal{V}}),$$

where Self-Attn is the self-attention module, Cross-Attn is the cross-attention module, $\boldsymbol{h}_t^s, \tilde{\boldsymbol{h}}_t^c, \tilde{\boldsymbol{h}}_t^p, \boldsymbol{h}_t \in \mathbb{R}^h$ are self-attended hidden state, observation-related hidden state, progression-related hidden state, and spatiotemporal-aware hidden state, respectively, $\boldsymbol{W}_\alpha \in \mathbb{R}^h, \boldsymbol{W}_{\mathcal{V}} \in \mathbb{R}^{|\mathcal{V}| \times h}$ are weight matrices and $b_\alpha \in \mathbb{R}, \boldsymbol{b}_{\mathcal{V}} \in \mathbb{R}^{|\mathcal{V}|}$ are the biases.

**Disease Progression Encoding.** As there are different relations between nodes, we adopt an $L$-layer Relational Graph Convolutional Network (R-GCN) (Schlichtkrull et al., 2018) to encode the disease progression graph, similar to Ji et al. (2020):

$$\boldsymbol{h}_{v_i}^{l+1} = \text{ReLU}\left(\frac{1}{c_i} \sum_{r_j \in R}^{r_j \in R} \sum_{v_k \in V} \boldsymbol{W}_{r_j}^l \boldsymbol{h}_{v_k}^l + \boldsymbol{W}_0^l \boldsymbol{h}_{v_i}^l\right),$$

where $c_i$ is the number of neighbors connected to the $i$-th node, $\boldsymbol{W}_{r_j}^l, \boldsymbol{W}_0^l \in \mathbb{R}^{h \times h}$ are learnable weight metrics, and $\boldsymbol{h}_{v_i}^l, \boldsymbol{h}_{v_i}^{l+1}, \boldsymbol{h}_{v_k}^l \in \mathbb{R}^h$ are hidden representations.

| Dataset | Model | NLG Metrics | | | | | | CE Metrics | | |
|---|---|---|---|---|---|---|---|---|---|---|
| | | B-1 | B-2 | B-3 | B-4 | MTR | R-L | P | R | $F_1$ |
| MIMIC -ABN | R2GEN | 0.290 | 0.157 | 0.093 | 0.061 | 0.105 | 0.208 | 0.266 | 0.320 | 0.272 |
| | R2GENCMN | 0.264 | 0.140 | 0.085 | 0.056 | 0.098 | 0.212 | 0.290 | 0.319 | 0.280 |
| | ORGAN | 0.314 | 0.180 | 0.114 | 0.078 | **0.120** | **0.234** | 0.271 | 0.342 | 0.293 |
| | RECAP (Ours) | **0.321** | **0.182** | **0.116** | **0.080** | **0.120** | 0.223 | **0.300** | **0.363** | **0.305** |
| MIMIC -CXR | R2GEN | 0.353 | 0.218 | 0.145 | 0.103 | 0.142 | 0.270 | 0.333 | 0.273 | 0.276 |
| | R2GENCMN | 0.353 | 0.218 | 0.148 | 0.106 | 0.142 | 0.278 | 0.344 | 0.275 | 0.278 |
| | $\mathcal{M}^2$TR | 0.378 | 0.232 | 0.154 | 0.107 | 0.145 | 0.272 | 0.240 | 0.428 | 0.308 |
| | KNOWMAT | 0.363 | 0.228 | 0.156 | 0.115 | – | 0.284 | **0.458** | 0.348 | 0.371 |
| | CMM-RL | 0.381 | 0.232 | 0.155 | 0.109 | 0.151 | 0.287 | 0.342 | 0.294 | 0.292 |
| | CMCA | 0.360 | 0.227 | 0.156 | 0.117 | 0.148 | 0.287 | 0.444 | 0.297 | 0.356 |
| | KiUT | 0.393 | 0.243 | 0.159 | 0.113 | 0.160 | 0.285 | 0.371 | 0.318 | 0.321 |
| | DCL | – | – | – | 0.109 | 0.150 | 0.284 | 0.471 | 0.352 | 0.373 |
| | METrans | 0.386 | 0.250 | 0.169 | 0.124 | 0.152 | 0.291 | 0.364 | 0.309 | 0.311 |
| | ORGAN | 0.407 | 0.256 | 0.172 | 0.123 | 0.162 | **0.293** | 0.416 | 0.418 | 0.385 |
| | RECAP (Ours) | **0.429** | **0.267** | **0.177** | **0.125** | **0.168** | 0.288 | 0.389 | **0.443** | **0.393** |

Table 1: Experimental Results of our model and baselines on the MIMIC-ABN and MIMIC-CXR datasets. The best results are in **boldface**, and the underlined are the second-best results. The experimental results on the MIMIC-ABN dataset are replicated based on their corresponding repositories.

**Precise Report Decoding via Progression Reasoning.** Inspired by Ji et al. (2020) and Mu and Li (2022), we devise a dynamic disease progression reasoning (PrR) mechanism to select observation-relevant attributes from the progression graph. The reasoning path of PrR is $o_i^c \xrightarrow{r_j} e_k$, where $r_j$ belongs to either three kinds of progression or $R_s$. Specifically, given $t$-th hidden representation $\boldsymbol{h}_t$, the observation representation $\boldsymbol{h}_{o_i}^L$, and the entity representation $\boldsymbol{h}_{e_k}^L$ of $e_k$, the progression score $\hat{ps}_t(e_k)$ of node $e_k$ is calculated as:

$$ps_t(e_k) = \frac{1}{|\mathcal{N}_{e_k}|} \sum_{(o_i, r_j) \in \mathcal{N}_{e_k}} \phi(\boldsymbol{h}_t^\mathsf{T} \boldsymbol{W}_{r_i}[\boldsymbol{h}_{o_i}^L; \boldsymbol{h}_{e_k}^L]),$$

$$\hat{ps}_t(e_k) = \gamma \cdot ps_t(e_k) + \phi(\boldsymbol{h}_t \boldsymbol{W}_s \boldsymbol{h}_{e_k}^L),$$

where $\phi$ is the Tangent function, $\gamma$ is the scale factor, $\mathcal{N}_{e_k}$ is the neighbor collection of $e_k$, and $\boldsymbol{W}_{r_i} \in \mathbb{R}^{h \times 2h}$ and $\boldsymbol{W}_s \in \mathbb{R}^{h \times h}$ are weight matrices for learning relation $r_i$ and self-connection, respectively. In the PrR mechanism, the relevant scores (i.e., $ps_t(e_k)$) of their connected observations are also included in $\hat{ps}_t(e_k)$ since $\boldsymbol{h}_t$ contains observation information, and higher relevant scores of these connected observations indicate a higher relevant score of $e_k$. Then, the distribution over all entities in $G$ is denoted as:

$$p_G(y_t) = \text{Softmax}(\hat{ps}_t(e_k)).$$

Finally, a soft gate $g_t = \sigma(\boldsymbol{W}_g \boldsymbol{h}_t + b_g)$ is adopted

to combine $p_\mathcal{V}(y_t)$ and $p_G(y_t)$ into $p(y_t)$:

$$p(y_t) = g_t \cdot p_\mathcal{V}(y_t) + (1 - g_t) \cdot p_G(y_t),$$

where $\boldsymbol{W}_g \in \mathbb{R}^h$ and $b_g \in \mathbb{R}$ are the weight matrix and bias, respectively.

**Training.** The generation process is optimized using the negative log-likelihood loss, given each token's probability $p(y_t)$ and the probability of $g_t$:

$$\mathcal{L}_{\text{NLL}} = - \sum_{t=1}^{T} \log p(y_t),$$

$$\mathcal{L}_g = - \sum_{t=1}^{T} [l_{g_t} \log g_t + (1 - l_{g_t}) \log(1 - g_t)],$$

where $l_{g_t}$ indicates $t$-th token appears in $G$. Finally, the loss of Stage 2 is $\mathcal{L}_{S2} = \mathcal{L}_{\text{NLL}} + \lambda \mathcal{L}_g$.

## 4 Experiments

### 4.1 Datasets

We use two benchmarks to evaluate our models, MIMIC-ABN[5] (Ni et al., 2020) and MIMIC-CXR[6] (Johnson et al., 2019). We provide other details of data preprocessing in Appendix A.3.

- MIMIC-CXR consists of 377,110 chest X-ray images and 227,827 reports from 63,478 patients. We adopt the settings of Chen et al. (2020).

---
[5] https://github.com/zzxslp/WCL
[6] https://physionet.org/content/mimic-cxr-jpg/2.0.0/

| Model | Sections | B-2 | CE-F$_1$ |
|---|---|---|---|
| R2GEN | *Find. & Imp.* | 0.212 | 0.148 |
| IFCC | *Findings* | 0.217 | 0.270 |
| CXR-RePaiR-Sel | *Impressions* | 0.050 | 0.274 |
| BioViL-T | *Impressions* | 0.159 | 0.348 |
| BioViL-T | *Find. & Imp.* | 0.213 | 0.359 |
| ORGAN | *Findings* | 0.267 | 0.385 |
| RECAP (Ours) | *Findings* | **0.265** | **0.393** |

Table 2: BLEU score and CheXbert score of our model and baselines on the MIMIC-CXR dataset. Results are cited from Bannur et al. (2023) and Hou et al. (2023).

- MIMIC-ABN is a modified version of MIMIC-CXR and only contains abnormal sentences. The original train/validation/test split of Ni et al. (2020) is 26,946/3,801/7,804 samples, respectively. To collect patients' historical information and avoid information leakage, we recover the data-split used in MIMIC-CXR according to the *subject_id*[7]. Finally, the data-split used in our experiments is 71,786/546/806 for train/validation/test sets, respectively.

## 4.2 Evaluation Metrics and Baselines

**NLG Metrics.** BLEU (Papineni et al., 2002), METEOR (Banerjee and Lavie, 2005), and ROUGE (Lin, 2004) are selected as the Natural Language Generation metrics (NLG Metrics), and we use the MS-COCO evaluation tool[8] to compute the results.
**CE Metrics.** For Clinical Efficacy (CE Metrics), CheXbert[9] (Smit et al., 2020) is adopted to label the generated reports compared with disease labels of the references. Besides, we use the temporal entity matching scores (TEM), proposed by Bannur et al. (2023), to evaluate how well the models generate progression-related information.
**Baselines.** For performance evaluation, we compare our model with the following state-of-the-art (SOTA) baselines: R2GEN (Chen et al., 2020), R2GENCMN (Chen et al., 2021), KNOWMAT (Yang et al., 2021), $\mathcal{M}^2$TR (Nooralahzadeh et al., 2021), CMM-RL (Qin and Song, 2022), CMCA (Song et al., 2022), CXR-RePaiR-Sel/2 (Endo et al., 2021), BioViL-T (Bannur et al., 2023), DCL (Li et al., 2023), METrans (Wang et al., 2023), KiUT (Huang et al., 2023), and ORGAN (Hou et al., 2023).

---

[7] *subject_id* is the anonymized identifier of a patient.
[8] https://github.com/tylin/coco-caption
[9] https://github.com/stanfordmlgroup/CheXbert

| Model | B-4 | R-L | CE-F$_1$ | TEM |
|---|---|---|---|---|
| CXR-RePaiR-2 | 0.021 | 0.143 | 0.281 | 0.125 |
| BioViL-NN | 0.037 | 0.200 | 0.283 | 0.111 |
| BioViL-T-NN | 0.045 | 0.205 | 0.290 | 0.130 |
| BioViL-AR | 0.075 | 0.279 | 0.293 | 0.138 |
| BioViL-T-AR | 0.092 | **0.296** | 0.317 | 0.175 |
| RECAP (Ours) | **0.118** | 0.279 | 0.400 | **0.304** |
| RECAP *w/o* OP | 0.093 | 0.260 | 0.256 | 0.203 |
| RECAP *w/o* Obs | 0.104 | 0.270 | 0.307 | 0.240 |
| RECAP *w/o* Pro | 0.103 | 0.266 | 0.395 | 0.269 |
| RECAP *w/o* PrR | 0.115 | 0.279 | **0.403** | 0.296 |

Table 3: Progression modeling performance of our model and baselines on the MIMIC-CXR dataset. The *-NN models use nearest neighbor search for report generation, and the *-AR models use autoregressive decoding, as indicated in Bannur et al. (2023).

## 4.3 Implementation Details

We use the ViT (Dosovitskiy et al., 2021), a vision transformer pretrained on ImageNet (Deng et al., 2009), as the visual encoder[10]. The maximum decoding step is set to 64/104 for MIMIC-ABN and MIMIC-CXR, respectively. $\gamma$ is set to 2 and $K$ is set to 30 for both datasets.

For model training, we adopt AdamW (Loshchilov and Hutter, 2019) as the optimizer. The layer number of the Transformer encoder and decoder are both set to 3, and the dimension of the hidden state is set to 768, which is the same as the one of ViT. The layer number $L$ of the R-GCN is set to 3. The learning rate is set to 5e-5 and 1e-4 for the pretrained ViT and the rest of the parameters, respectively. The learning rate decreases from the initial learning rate to 0 with a linear scheduler. The dropout rate is set to 0.1, the batch size is set to 32, and $\lambda$ is set to 0.5. We select the best checkpoints based on the BLEU-4 on the validation set. Our model has 160.05M trainable parameters, and the implementations are based on the HuggingFace's *Transformers* (Wolf et al., 2020). We implement our models on an NVIDIA-3090 GTX GPU with mixed precision. Other details of implementation (e.g., Stage 1 training) can be found in Appendix A.3.

## 5 Results

### 5.1 Quantitative Analysis

**NLG Results.** The NLG results of two datasets are listed on the left side of Table 1 and Table 2. As we can see from Table 1, RECAP achieves the best

---

[10] The model card is "google/vit-base-patch16-224-in21k."

| Dataset | Model | NLG Metrics | | | | | | CE Metrics | | |
|---|---|---|---|---|---|---|---|---|---|---|
| | | B-1 | B-2 | B-3 | B-4 | MTR | R-L | P | R | $F_1$ |
| MIMIC-ABN | RECAP | 0.321 | 0.182 | 0.116 | 0.080 | 0.120 | 0.223 | 0.300 | 0.363 | 0.305 |
| | RECAP *w/o* OP | 0.303 | 0.170 | 0.109 | 0.074 | 0.113 | 0.227 | 0.289 | 0.300 | 0.280 |
| | RECAP *w/o* Obs | 0.302 | 0.174 | 0.114 | 0.079 | 0.114 | 0.231 | 0.341 | 0.314 | 0.282 |
| | RECAP *w/o* Pro | 0.306 | 0.169 | 0.107 | 0.072 | 0.114 | 0.220 | 0.298 | 0.361 | 0.298 |
| | RECAP *w/o* PrR | 0.320 | 0.180 | 0.115 | 0.079 | 0.120 | 0.224 | 0.295 | 0.365 | 0.301 |
| MIMIC-CXR | RECAP | 0.429 | 0.267 | 0.177 | 0.125 | 0.168 | 0.288 | 0.389 | 0.443 | 0.393 |
| | RECAP *w/o* OP | 0.350 | 0.219 | 0.150 | 0.109 | 0.140 | 0.278 | 0.356 | 0.259 | 0.266 |
| | RECAP *w/o* Obs | 0.356 | 0.224 | 0.153 | 0.113 | 0.144 | 0.283 | 0.464 | 0.281 | 0.296 |
| | RECAP *w/o* Pro | 0.402 | 0.245 | 0.161 | 0.112 | 0.157 | 0.278 | 0.379 | 0.433 | 0.386 |
| | RECAP *w/o* PrR | 0.415 | 0.257 | 0.171 | 0.119 | 0.164 | 0.285 | 0.381 | 0.443 | 0.391 |

Table 4: Ablation results of our model and its variants. RECAP *w/o* OP is the standard Transformer model, *w/o* Obs stands for without observation, and *w/o* Pro stands for without progression.

performance compared with other SOTA models and outperforms other baselines substantially on both datasets.

**Clinical Efficacy Results.** The clinical efficacy results are shown on the right side of Table 1. RECAP achieves SOTA performance on $F_1$ score, leading to a 1.2% improvement over the best baseline (i.e., ORGAN) on the MIMIC-ABN dataset. Similarly, on the MIMIC-CXR dataset, our model achieves a score of 0.393, increasing by 0.8% compared with the second-best. This demonstrates that RECAP can generate better clinically accurate reports.

**Temporal-related Results.** Since there are only 10% follow-up-visits records in the MIMIC-ABN dataset, we mainly focus on analyzing the MIMIC-CXR dataset, as shown in Table 3 and Table 6. RECAP achieves the best performance on BLEU-4, TEM. In terms of the clinical $F_1$, RECAP *w/o* PrR outperforms other baselines. This indicates that historical records are necessary for generating follow-up reports.

**Ablation Results.** We perform ablation analysis, and the ablation results are listed in Table 4. We also list the ablation results on progression modeling in Table 6. There are four variants: (1) RECAP *w/o* OP (i.e., a standard Transformer model, removing spatiotemporal information), (2) RECAP *w/o* Obs (i.e., without observation), (3) RECAP *w/o* Pro (i.e., without progression), and (4) RECAP *w/o* PrR, which does not adopt the disease progression reasoning mechanism.

As we can see from Table 4, without the spatiotemporal information (i.e., variant 1), the performances drop significantly on both datasets, which indicates the necessity of spatiotemporal modeling in free-text report generation. In addition, compared with variant 1, the performance of RECAP *w/o* Obs increases substantially on the MIMIC-CXR dataset, which demonstrates the importance of historical records in assessing the current conditions of patients. In terms of CE metrics, learning from the observation information boosts the performance of RECAP drastically, with an improvement of 12%. In addition, the performance of RECAP increases compared with variant *w/o* PrR. This indicates that PrR can help generate precise and accurate reports.

## 5.2 Qualitative Analysis

**Case Study.** We conduct a detailed case study on how RECAP generates precise and accurate attributes of a given radiograph in Figure 3. RECAP successfully generates six observations, including five abnormal observations. Regarding attribute modeling, our model can generate the precise description "*the lungs are clear without focal consolidation*", which also appears in the reference, while RECAP *w/o* OP can not generate relevant descriptions. This indicates that spatiotemporal information plays a vital role in the generation process. Additionally, RECAP can learn to compare with the historical records (e.g., *mediastinal contours are stable and remarkable*) so as to precisely measure the observations.

**Error Analysis.** We depict error analysis to provide more insights, as shown in Figure 4. There are two major errors, which are false-positive observations (i.e., Positive Lung Opacity and Positive Pleural Effusion) and false-negative observations (i.e., Negative Cardiomegaly). Improving the performance of observation prediction could be an important direction in enhancing the quality of gener-

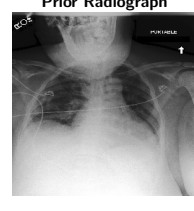

**Prior Radiograph**

**Current Radiograph**
**Progression: Stable**

① Enlarged Card./NEG
② Cardiomegaly/POS
③ Edema/NEG
④ Consolidation/NEG
⑤ Pneumothorax/NEG
⑥ Pleural Effusion/NEG

**Prior Report:** low lung volumes are present. ② this accentuates the size of the cardiac silhouette which is likely mildly enlarged. ① mediastinal and hilar contours are likely within normal limits. a right brachiocephalic venous stent is re-demonstrated. there is crowding of the bronchovascular structures with probable ③ mild pulmonary vascular congestion. ⑥ no pleural effusion or ⑤ pneumothorax is identified.

**Reference:** ap and lateral views of the chest. the lungs are ④ clear of consolidation or ⑥ effusion. ② the cardiac silhouette is **enlarged** but **unchanged**. no acute osseous abnormality is detected. right brachiocephalic venous stent is again noted.

**Ours:** the lungs are **clear** without focal consolidation. no pleural effusion or pneumothorax is seen. the cardiac silhouette is **top-normal** to **mildly enlarged**. mediastinal contours are **stable** and **unremarkable**. there is no pulmonary edema.

**Ours *w/o* OP:** ap upright and lateral views of the chest provided. lung volumes are **low** limiting assessment. allowing for this there is ④ no focal consolidation ⑥ effusion or ⑤ pneumothorax. ① the cardiomediastinal silhouette is **normal**. imaged osseous structures are **intact**. no free air below the right hemidiaphragm is seen.

Figure 3: Case study of a follow-up-visit sample, given its prior radiograph and prior report. Attributes of observations in reports are highlighted in **boldface**, and spans with colors in reports indicate mentions of observations.

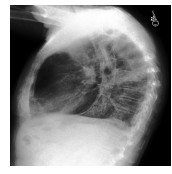

① Enlarged Card./NEG
② Cardiomegaly/F.NEG
③ Lung Opacity/F.POS
④ Pneumothorax/NEG
⑤ Effusion/F.POS

**Reference:** there is no new consolidation. right lower lobe pneumonia that was present in prior exams has significantly improved. esophageal stent is in unchanged position. there is no pneumomediastinum or pneumothorax. there is no pleural effusion. mediastinal and cardiac contours are stable.

**Ours:** ② the heart size is normal. the hilar and mediastinal contours are within normal limits. there is no pneumothorax. again seen is a ⑤ small right pleural effusion. the visualized osseous structures are unremarkable. there has been interval improvement of the ③ right basilar opacity.

Figure 4: Error case generated by RECAP. The span and the spans denote false negative observation and false positive observation, respectively.

ated reports. In addition, although RECAP aims to model precise attributes of observations presented in the radiograph, it still can not cover all the cases. This might be alleviated by incorporating external knowledge.

## 6 Related Work

### 6.1 Medical Report Generation

Medical report generation(Jing et al., 2018; Li et al., 2018), as one kind of image captioning (Vinyals et al., 2015; Rennie et al., 2017; Lu et al., 2017; Anderson et al., 2018), has received increasing attention from the research community. Some works focus on recording key information of the generation process via memory mechanism (Chen et al., 2020, 2021; Qin and Song, 2022; Wang et al., 2023). In addition, Liu et al. (2021c) proposed to utilize contrastive learning to distill information. Liu et al. (2021a) proposed to use curriculum learning to enhance the performance and Liu et al. (2021b) proposed to explore posterior and prior knowledge for report generation. Yang et al. (2021); Li et al. (2023); Huang et al. (2023) proposed to utilize the external knowledge graph (i.e., RadGraph (Jain et al., 2021)) for report generation.

Other works focused on improving the clinical accuracy and faithfulness of the generated reports. Liu et al. (2019a); Lovelace and Mortazavi (2020); Miura et al. (2021); Nishino et al. (2022); Delbrouck et al. (2022) designed various kinds of rewards (e.g., entity matching score) to improve clinical accuracy via reinforcement learning. Tanida et al. (2023) proposed an explainable framework for report generation that could identify the abnormal areas of a given radiograph. Hou et al. (2023) proposed to combine both textual plans and radiographs to maintain clinical consistency. Additionally, Ramesh et al. (2022) and Bannur et al. (2023) focus on handling the temporal structure in radiology report generation, either removing the prior or learning from the historical records.

### 6.2 Graph Reasoning for Text Generation

Graph reasoning for text generation (Liu et al., 2019b; Tuan et al., 2022) tries to identify relevant knowledge from graphs and incorporate it into generated text sequences. Huang et al. (2020) proposed to construct a knowledge graph from the input document and utilize it to enhance the performance of abstractive summarization. Ji et al. (2020) proposed to incorporate commonsense knowledge for

language generation via multi-hop reasoning. Mu and Li (2022) proposed to combine both event-level and token-level from the knowledge graph to improve the performance.

# 7 Conclusion

In this paper, we propose RECAP, which can capture both spatial and temporal information for generating precise and accurate radiology reports. To achieve precise attribute modeling in the generation process, we construct a disease progression graph containing both observations and fined-grained attributes which quantify the severity of diseases and devise a dynamic disease progression reasoning (PrR) mechanism to select observation-relevant attributes. Experimental results demonstrate the effectiveness of our proposed model in terms of generating precise and accurate radiology reports.

## Limitations

Our proposed two-stage framework requires pre-defined observations and progressions for training, which may not be available for other types of radiographs. In addition, the outputs of Stage 1 are the prerequisite inputs of Stage 2, and thus, our framework may suffer from error propagation. Finally, although prior information is important in generating precise and accurate free-text reports, historical records are not always available, even in the two benchmark datasets. Our framework will still generate misleading free-text reports, conditioning on non-existent priors, as indicated in Ramesh et al. (2022). This might be mitigated through rule-based removal operations.

## Ethics Statement

The MIMIC-ABN(Ni et al., 2020) and MIMIC-CXR (Johnson et al., 2019) datasets are publicly available benchmarks and have been automatically de-identified to protect patient privacy. Although our model improves the factual accuracy of generated reports, its performance still lags behind the practical deployment. The outputs of our model may contain false observations and diagnoses due to systematic biases. In this regard, we strongly urge the users to examine the generated output in real-world applications cautiously.

## Acknolwedgments

This work was supported in part by General Program of National Natural Science Foundation of China (Grant No. 82272086, 62076212), Guangdong Provincial Department of Education (Grant No. 2020ZDZX3043), Shenzhen Natural Science Fund (JCYJ20200109140820699 and the Stable Support Plan Program 20200925174052004), and the Research Grants Council of Hong Kong (15207920, 15207821, 15207122).

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

# A Appendix

## A.1 Observation and Progression Statitics

There are 14 observations: No Finding, Enlarged Cardiomediastinum, Cardiomegaly, Lung Lesion, Lung Opacity, Edema, Consolidation, Pneumonia, Atelectasis, Pneumothorax, Pleural Effusion, Pleural Other, Fracture, and Support Devices. Table

5 lists the observation distributions annotated by CheXbert(Smit et al., 2020) in the train/valid/test split of two benchmarks and Table 7 shows the progression distributions in our experiments.

| #Observation | MIMIC-ABN | MIMIC-CXR |
|---|---|---|
| *No Finding*/POS | 5002/32/22 | 64,677/514/229 |
| *No Finding*/NEG | 66,784/514/784 | 206,133/1,616/3,629 |
| *Cardiomegaly*/POS | 16,312/118/244 | 70,561/514/1,602 |
| *Cardiomegaly*/NEG | 804/4/8 | 85,448/714/801 |
| *Pleural Effusion*/POS | 10,502/80/186 | 56,972/477/1,379 |
| *Pleural Effusion*/NEG | 1,948/18/24 | 170,989/1,310/1,763 |
| *Pneumothorax*/POS | 1,452/24/4 | 8,707/62/106 |
| *Pneumothorax*/NEG | 1,792/10/26 | 190,356/1,495/2,338 |
| *Enlarged Card.*/POS | 5,202/40/90 | 49,806/413/1,140 |
| *Enlarged Card.*/NEG | 1,194/10/14 | 129,360/1,006/868 |
| *Consolidation*/POS | 4,104/36/96 | 14,449/119/384 |
| *Consolidation*/NEG | 3,334/20/34 | 97,197/788/964 |
| *Lung Opacity*/POS | 22,598/166/356 | 67,714/497/1,448 |
| *Lung Opacity*/NEG | 748/10/4 | 8,157/73/125 |
| *Fracture*/POS | 4,458/32/76 | 11,070/59/232 |
| *Fracture*/NEG | 330/0/0 | 9,632/72/53 |
| *Lung Lesion*/POS | 5,612/54/112 | 11,717/123/300 |
| *Lung Lesion*/NEG | 120/2/2 | 1,972/21/11 |
| *Edema*/POS | 8,704/76/168 | 33,034/257/899 |
| *Edema*/NEG | 1,898/16/32 | 51,639/409/669 |
| *Atelectasis*/POS | 19,132/134/220 | 68,273/515/1,210 |
| *Atelectasis*/NEG | 116/2/0 | 563/5/9 |
| *Support Devices*/POS | 9,886/58/196 | 60,455/450/1,358 |
| *Support Devices*/NEG | 394/0/10 | 1,081/7/11 |
| *Pneumonia*/POS | 17,826/138/260 | 23,945/184/503 |
| *Pneumonia*/NEG | 3,226/22/34 | 21,976/165/411 |
| *Pleural Other*/POS | 2,850/30/62 | 7,296/70/184 |
| *Pleural Other*/NEG | 8/0/0 | 63/0/0 |

Table 5: Observation distribution in train/valid/test split of two benchmarks. *Enlarged Card.* refers to *Enlarged Cardiomediastinum.*

## A.2 Spatial and Temporal Entity

Here are some of the spatial entities: healed, fractured, healing, nondisplaced, top, size, heart, normal, mediastinum, widening, contour, widened, consolidative, collapse, underlying, developing, fibrosis, thickening, biapical, blunting, indistinctness, asymmetrical, haziness, asymmetric, layering, subpulmonic, thoracentesis, trace, small, adjacent, tiny, atypical, developing, supervening, multifocal, correct, superimposed, patchy, and borderline. For temporal entities, we use the same settings of Bannur et al. (2023), which are: bigger, change, cleared, constant, decrease, decreased, decreasing, elevated, elevation, enlarged, enlargement, enlarging, expanded, greater, growing, improved, improvement, improving, increase, increased, increasing, larger, new, persistence, persistent, persisting, progression, progressive, reduced, removal, resolution, resolved, resolving, smaller, stability, stable, stably, unchanged, unfolded, worse, worsen, wors-

| Dataset | Model | NLG Metrics | | | | | | CE Metrics | | |
|---|---|---|---|---|---|---|---|---|---|---|
| | | **B-1** | **B-2** | **B-3** | **B-4** | **MTR** | **R-L** | **P** | **R** | **F$_1$** |
| | | | | | *w.* Historical Record $D^p$ | | | | | |
| MIMIC -ABN | RECAP | 0.327 | 0.183 | 0.117 | 0.081 | 0.124 | 0.227 | 0.274 | 0.372 | 0.297 |
| | RECAP *w/o* OP | 0.300 | 0.164 | 0.106 | 0.072 | 0.110 | 0.217 | 0.281 | 0.274 | 0.257 |
| | RECAP *w/o* Obs | 0.306 | 0.173 | 0.110 | 0.076 | 0.114 | 0.233 | 0.270 | 0.288 | 0.259 |
| | RECAP *w/o* Pro | 0.295 | 0.158 | 0.099 | 0.070 | 0.109 | 0.209 | 0.249 | 0.361 | 0.278 |
| | RECAP *w/o* PrR | 0.320 | 0.177 | 0.112 | 0.076 | 0.121 | 0.218 | 0.266 | 0.377 | 0.292 |
| MIMIC -CXR | RECAP | 0.423 | 0.260 | 0.170 | 0.118 | 0.169 | 0.279 | 0.387 | 0.457 | 0.400 |
| | RECAP *w/o* OP | 0.321 | 0.196 | 0.131 | 0.093 | 0.130 | 0.260 | 0.350 | 0.238 | 0.256 |
| | RECAP *w/o* Obs | 0.347 | 0.213 | 0.144 | 0.104 | 0.141 | 0.270 | 0.465 | 0.293 | 0.307 |
| | RECAP *w/o* Pro | 0.396 | 0.236 | 0.151 | 0.103 | 0.153 | 0.266 | 0.383 | 0.447 | 0.395 |
| | RECAP *w/o* PrR | 0.420 | 0.257 | 0.168 | 0.115 | 0.166 | 0.279 | 0.386 | 0.459 | 0.403 |
| | | | | | *w/o* Historical Record $D^p$ | | | | | |
| MIMIC -ABN | RECAP | 0.319 | 0.182 | 0.116 | 0.080 | 0.120 | 0.223 | 0.306 | 0.360 | 0.306 |
| | RECAP *w/o* OP | 0.303 | 0.171 | 0.109 | 0.074 | 0.110 | 0.217 | 0.299 | 0.302 | 0.283 |
| | RECAP *w/o* Obs | 0.301 | 0.174 | 0.114 | 0.079 | 0.114 | 0.231 | 0.353 | 0.313 | 0.282 |
| | RECAP *w/o* Pro | 0.309 | 0.171 | 0.109 | 0.073 | 0.115 | 0.222 | 0.314 | 0.360 | 0.302 |
| | RECAP *w/o* PrR | 0.320 | 0.181 | 0.116 | 0.079 | 0.120 | 0.225 | 0.299 | 0.362 | 0.302 |
| MIMIC -CXR | RECAP | 0.427 | 0.268 | 0.180 | 0.128 | 0.168 | 0.294 | 0.378 | 0.417 | 0.374 |
| | RECAP *w/o* OP | 0.371 | 0.236 | 0.164 | 0.121 | 0.130 | 0.260 | 0.357 | 0.259 | 0.268 |
| | RECAP *w/o* Obs | 0.363 | 0.231 | 0.161 | 0.119 | 0.146 | 0.291 | 0.415 | 0.262 | 0.277 |
| | RECAP *w/o* Pro | 0.406 | 0.251 | 0.151 | 0.103 | 0.153 | 0.266 | 0.364 | 0.405 | 0.365 |
| | RECAP *w/o* PrR | 0.412 | 0.257 | 0.172 | 0.122 | 0.163 | 0.289 | 0.364 | 0.414 | 0.368 |

Table 6: Ablation results of our model and its variants on progression modeling. RECAP *w/o* OP is the standard Transformer model, *w/o* Obs stands for without observation, and *w/o* Pro stands for without progression.

| #Progression | MIMIC-ABN | MIMIC-CXR |
|---|---|---|
| Better | 929/2/19 | 14,790/110/345 |
| Worse | 1,219/6/30 | 18,083/163/431 |
| Stable | 4,114/31/99 | 41,721/334/1,085 |
| Total | 6,440/48/137 | 64,498/535/1,566 |
| Ratio | 9%/8.8%/17% | 24%/25.1%/40.6% |

Table 7: Progression distribution in train/valid/test split of two benchmarks.

ened, worsening and unaltered.

### A.3 Other Implementation Details

**Data Preprocessing.** We adopt the preprocessing setup used in Chen et al. (2020), and the minimum count of each token is set to 3/10 for MIMIC-ABN/MIMIC-CXR, respectively. Other tokens are replaced with a special token [UNK].

**Implementation Details of Stage 1 Training.** Table 8 shows the hyperparameters used in Stage 1 training for two datasets. Note that $l_{d_i}$ is the weight for observation detection, and the weights of observation classification and progression classification are both set to 1. In addition, two data augmentation methods are used during training. Specifically,

we first resize an input image to $256 \times 256$, and then the image is randomly cropped to $224 \times 224$, and finally, we flip the image horizontally with a probability of 0.5. We select the best checkpoint based on the Macro-F$_1$ of abnormal observations at this stage.

| Hyperparameter | MIMIC-ABN | MIMIC-CXR |
|---|---|---|
| Training Epoch | 10 | 5 |
| Dropout Rate | 0.1 | 0.1 |
| Learning Rate | $1e-4$ | $1e-4$ |
| Batch Size | $\{64, \mathbf{128}\}$ | $\{64, \mathbf{128}\}$ |
| Sample Weight ($\alpha_d$) | $\{1, 2, \mathbf{3}\}$ | $\{1, 2, \mathbf{3}\}$ |

Table 8: Selected hyperparameters of Stage 1 training. The final hyperparameters in **boldface** are tuned on the validation set and others are set empirically.

**Implementation Details of Stage 2 Training.** As the variant *w/o* OP and the variant *w/o* Obs in Table 4 are not trained in Stage 1, they are trained with more epochs (i.e., 10 epochs).

### A.4 Other Experimental Results

We show experimental results of observation prediction and progression prediction during Stage 1

training in Table 9 and Table 10, respectively.

| Dataset | D-$F_1$ | A-$F_1$ | N-$F_1$ |
|---|---|---|---|
| MIMIC-ABN | 0.539 | 0.355 | 0.426 |
| MIMIC-CXR | 0.686 | 0.428 | 0.759 |

Table 9: Experimental results of observation prediction after Stage 1 training. D-$F_1$, A-$F_1$, and N-$F_1$ denote the $F_1$ of observation detection, abnormal observation prediction, and normal observation prediction, respectively.

| Dataset | Better | Worse | Stable | Macro |
|---|---|---|---|---|
| MIMIC-ABN | 0.286 | 0.468 | 0.934 | 0.563 |
| MIMIC-CXR | 0.389 | 0.455 | 0.896 | 0.580 |

Table 10: Experimental results of progression prediction ($F_1$) after Stage 1 training.

| Observation | P | R | $F_1$ |
|---|---|---|---|
| Enlarged Card. | 0.323 | 0.589 | 0.417 |
| Cardiomegaly | 0.585 | 0.836 | 0.689 |
| Lung Opacity | 0.489 | 0.499 | 0.494 |
| Lung Lesion | 0.265 | 0.044 | 0.075 |
| Edema | 0.562 | 0.587 | 0.574 |
| Consolidation | 0.285 | 0.233 | 0.256 |
| Pneumonia | 0.242 | 0.444 | 0.313 |
| Atelectasis | 0.426 | 0.800 | 0.556 |
| Pneumothorax | 0.265 | 0.167 | 0.205 |
| Pleural Effusion | 0.691 | 0.781 | 0.728 |
| Pleural Other | 0.184 | 0.050 | 0.078 |
| Fracture | 0.155 | 0.081 | 0.107 |
| Support Devices | 0.720 | 0.660 | 0.689 |
| No Finding | 0.265 | 0.429 | 0.327 |
| Macro Average | 0.389 | 0.443 | 0.393 |

Table 11: Experimental results of each observation after Stage 2 training.