# OpenReview forum: "RECAP: Towards Precise Radiology Report Generation via Dynamic Disease Progression Reasoning"
_EMNLP/2023/Conference — EMNLP 2023 Findings_

### Official Review · Reviewer_7aGE · 2023-08-03

**Soundness:** 2

**Excitement:**

2: Mediocre: This paper makes marginal contributions (vs non-contemporaneous work), so I would rather not see it in the conference.

**Missing References:**

DeltaNet: Conditional medical report generation for COVID-19 diagnosis, Coling 2022

**Paper Topic And Main Contributions:**

In this paper, the authors target the medical report generation problem. Different from existing works, the authors propose to include historical reports in modeling and try to mine the disease progresses by comparing the historical reports with current input. The derived progressions are further used to improve report generation. The authors evaluate the proposed approach on the public MIMIC data set.

**Questions For The Authors:**

See the reason to reject section

**Reasons To Accept:**

1. Medical report generation is a critical clinical task which can relieve radiologists from the heavy workload.

2. Using historical reports and mining dissease progresses are useful for report generation.

**Reasons To Reject:**

1. The proposed approach fails to outperform existing works. For example, in Table 1, the B-4 of proposed approach is lower than the basic baseline ViT-transformer on MIMIC-ABN. Why the ViT-transformer is not evaluated on MIMIC-CXR data set.
2. What if the patients are the first time visitors without historical reports. The authors need to evaluate the proposed approach on new patients and old patients respectively.
3. The experiment setting is not fair. For the proposed approach, the historical reports of patients are used to generate reports for current input. While these data are unseen to baseline works. These historical reports should be added into the training data set of baselines.
4. One existing work which also includes historical reports in modeling should be referenced and discussed.
DeltaNet: Conditional medical report generation for COVID-19 diagnosis, Coling 2022.
5. Since the proposed approach targets to mine the progress of diseases to generate better results. Such intermediate results (the disease progress) should be evaluated in experiments.
6. The IU data set should be included in experiments.

**Reproducibility:**

4: Could mostly reproduce the results, but there may be some variation because of sample variance or minor variations in their interpretation of the protocol or method.

**Reviewer Confidence:**

4: Quite sure. I tried to check the important points carefully. It's unlikely, though conceivable, that I missed something that should affect my ratings.

---

> ### Author Rebuttal · Authors · 2023-08-27
>
> Thanks for R3’s review. After carefully reading R3’s comments, we find that R3 might have some misunderstanding about our experimental setting and might have overlooked some parts of our experiment section. More specific responses to R3’s concerns are as follows.
>
> $\textbf{Q1. Fails to outperform existing works:}$ The major objective of this paper is to generate precise and accurate radiology reports, so the CE metrics, which measure clinical accuracy, are our primary focus. As for the NLG metrics, they mainly calculate the similarity between the generations and the references in terms of n-gram accuracy. They might reflect the language quality of the generated reports to some extent, but they do not guarantee accuracy and reliability, which is the most critical concern in the medical field. That being said, our method still achieves competitive results on these dimensions. R3 mentioned that the B-4 of our method is lower than the one of ViT-Transformer, but in fact, there is only a 0.001 performance gap between them. Since there have already been over 10 baselines on the MIMIC-CXR dataset, including the most state-of-the-art ones, we do not put ViT-Transformer in the table.
>
> $\textbf{Q2. What if the patients are first-time visitors? The authors need to evaluate the proposed approach on new patients and old patients, respectively:}$ R3 might skip reading some important parts of the experiment section. In fact, we do provide the related experimental results on temporal modeling (i.e., results of “old patient”) in Table 3 and Table 7. The TEM score reflects how these models can generate progression-related information.
>
> $\textbf{Q3. Fairness of the experiment setting:}$ R3 might have some misunderstanding about our experimental setting. We want to point out that all the samples are included in the MIMIC-CXR and MIMIC-ABN datasets. Previous baselines treated and learned all the samples (including historical records) independently, and it does not mean that the historical records are unseen to these baselines. In other words, all the samples are seen to the baselines. Instead,  the overall setting is fair and proper, because the relations between consecutive studies play a vital role in generating precise and accurate reports, and again, this meets the objective of our paper.
>
> $\textbf{Q4. Why not include one more baseline (DeltaNet) published at COLING 2022:}$ We will consider including DeltaNet for discussion in the next version of our paper. Nonetheless, we want to point out that our current version has already considered over 10 baseline methods, including the most state-of-the-art ones published at top conferences, so we believe the comparisons in our experiments are already very sufficient. Moreover, the method mentioned by R3, which focuses on leveraging historical records or retrieved samples as contextual information for better COVID-19 report generation, is not closely related to our work that focuses on precise and accurate radiology report generation.
>
> $\textbf{Q5. Intermediate results should be evaluated:}$ We will include more qualitative analysis in the later version.
>
> $\textbf{Q6. Should include the IU dataset:}$ The IU dataset is not a suitable testbed for our method. The reasons are as follows: (1) The IU dataset does not contain historical records of patients, and (2) this dataset is too small for training a report generation system and for evaluating the performance of a model. In our current paper, we have conducted experiments on two commonly-sued datasets, which have already provided sufficient experimental results.

---

### Official Review · Reviewer_6ocQ · 2023-08-04

**Soundness:** 4

**Excitement:**

3: Ambivalent: It has merits (e.g., it reports state-of-the-art results, the idea is nice), but there are key weaknesses (e.g., it describes incremental work), and it can significantly benefit from another round of revision. However, I won't object to accepting it if my co-reviewers champion it.

**Missing References:**

In the related work its seems:
Improving the Factual Correctness of Radiology Report Generation with Semantic Rewards Jean-Benoit Delbrouck | Pierre Chambon | Christian Bluethgen | Emily Tsai | Omar Almusa | Curtis Langlotz, EMNLP22
falls into the category of "Other works focus on improving the clinical accuracy of the generated reports"


**Paper Topic And Main Contributions:**

- Authors propose RECAP, which can capture both spatial and temporal information for generating precise and accurate free-text reports.
- Authors construct a disease progression graph containing both observations and fine-grained attributes which quantify the severity of diseases. Then, we devise a dynamic disease progression reasoning (PrR) mechanism to select observation/progression-relevant attributes.
- Authors conduct extensive experiments on two publicly available benchmarks, and experimental results demonstrate the effectiveness of our model in generating precise and accurate radiology reports.

For the experiments, authors conduct their research on two datasets with an excellent comparison to prior work.

**Questions For The Authors:**

- Is there a reason you didnt use the bertscore?
- How does your model scores on F1Radgraph? [1]
- It seems like the Progression Graph is used as input to the model. Could you please clarify which radgraph or label information (or any information originating from the reports) is predicted and which information is used as input to the model? Or more broadly, could you clarify which information you need at test-time to generate a radiology report?

[1] Improving the Factual Correctness of Radiology Report Generation with Semantic Rewards Jean-Benoit Delbrouck | Pierre Chambon | Christian Bluethgen | Emily Tsai | Omar Almusa | Curtis Langlotz, EMNLP22

**Reasons To Accept:**

- The Spatial/Temporal component of radiology reports has long been overlooked. This work is a great step toward this direction.
- Extensive comparison to prior work
- Use of RadGraph to compute a semantic graph and progression
- Substantial improvements on CE Metrics.
- Results are reported in a wide range of metrics



**Reasons To Reject:**

- The model is quiet complex and difficult to understand
- It seems some information from the ground-truth reports are used as input to the model (at test-time). I might be wrong about this one (see questions for the authors)

**Reproducibility:**

3: Could reproduce the results with some difficulty. The settings of parameters are underspecified or subjectively determined; the training/evaluation data are not widely available.

**Reviewer Confidence:**

4: Quite sure. I tried to check the important points carefully. It's unlikely, though conceivable, that I missed something that should affect my ratings.

---

> ### Author Rebuttal · Authors · 2023-08-27
>
> We genuinely appreciate the time and effort that you have dedicated to providing valuable feedback on our manuscript, especially on the aspect of spatial/temporal modeling. We understand the concerns about the model's complexity and will improve the presentation for better comprehension. We will include the EMNLP'22 paper for discussion in the related works section.
>
> We want to emphasize that there is no information/label leakage in our framework. The progression graph is built purely based on the training corpus (i.e., statistical information). Here are the answers to the questions:
>
> For Q1: We follow previous works and only adopt word-overlap metrics (e.g., BLEU) for evaluation. A typical radiology report consists of several short sentences with each corresponding to one or two observations. Instead of assessing the overall quality of it, which the BERTScore will surely do, we pay more attention to phrases (e.g., n-grams). In this case, we do not adopt it for evaluation. However, we notice that BERT could be a good indication of report quality and will consider including such a score for better comparison.
>
> For Q2: For the MIMIC-CXR dataset, our model scores ~29% F1 on RadGraph, which demonstrates a substantial improvement over the baselines trained with the NLL objective presented in [1]. Actually, we conduct all our experiments with evaluation based on RadGraph and will include more details in the updated version.
>
> For Q3: Basically, at test time, the prerequisites are:
>
> (1) historical records (if exist, i.e., prior image and prior report),
>
> (2) the current image, and
>
> (3) the progression graph, or statistical information extracted from the training corpus (i.e., entities from RadGraph, and triples mined using PMI).
>
> The progression (sub-)graph is built on the fly after we predict the observations and progressions based on the images. There is no other label from the ground truth in our case. Here are some steps in generating a report:
>
> (1) observation/progression prediction based on historical records + the current image,
>
> (2) progression sub-graph retrieval based on the result of step (1), and
>
> (3) report generation based on the results of steps (1) and (2).

---

### Official Review · Reviewer_Uawd · 2023-08-05

**Soundness:** 4

**Excitement:**

4: Strong: This paper deepens the understanding of some phenomenon or lowers the barriers to an existing research direction.

**Paper Topic And Main Contributions:**

The paper proposes a method RECAP to leverage the data of the last visit and current visit of a patient to generate radiology reports so that the disease progression can be considered in the modelling. RECAP has two stages of training. The model was evaluated on MIMIC-CXR and MIMIC-ABN. Both ablation studies and comparison studies show promising results of the proposed method.

**Questions For The Authors:**

Line 211-212, are there gold labels available for training this task?

**Reasons To Accept:**

Good motivation for using longitudinal data with disease progression information to generate radiology reports. This can potentially be used to generate other types of reports or other clinical NLP/NLG tasks.

The comparison experiments and ablation studies show promising results of the proposed method, especially the benefit of considering information from the last visit of a patient. The examples are also insightful.

The paper is overall well-written.

**Reasons To Reject:**

The experiments are restricted to the MIMIC dataset only. It is unclear how this method may perform on a different dataset or different types of notes.

The model only considers the most recent last visit of the patient. What if more historical visits are considered?

**Reproducibility:**

4: Could mostly reproduce the results, but there may be some variation because of sample variance or minor variations in their interpretation of the protocol or method.

**Reviewer Confidence:**

4: Quite sure. I tried to check the important points carefully. It's unlikely, though conceivable, that I missed something that should affect my ratings.

**Typos Grammar Style And Presentation Improvements:**

It can be insightful to demonstrate evaluation metrics on a subset of MIMIC-CXR, where patients have recent visits, i.e. $D^p \neq \emptyset$. This may help demonstrate the advantage of using disease progression information better.

---

> ### Author Rebuttal · Authors · 2023-08-27
>
> We sincerely appreciate your valuable feedback on our work regarding the motivation and effectiveness of introducing longitudinal information.
>
> Historical records are difficult to collect. Patients are usually first diagnosed with some diseases, and then they will be asked to attend a follow-up check to track their conditions. However, patients with abnormal conditions are the minority in most cases. As a result, studies of patients with historical records become even rarer (i.e., data sparsity). In addition, introducing all the historical records would be time/space-inefficient regarding the performance gain (i.e., efficiency). Considering these two factors, we only use MIMIC datasets for experiments and introduce the last visit in our framework, but we think this could be a good chance for better visualization of progression, and we will conduct related experiments later for a better understanding of such temporal information.
>
> Because of the data sparsity, there are only a few datasets available for progression modeling and the MIMIC collection is one of them. We hope our work can encourage more research related to this topic and motivate the research community to collect and release other kinds of notes and data.
>
> Regarding the performance of patients who have recent visits, there are two tables (i.e., Table 3 on page 6 and Table 7 on page 13) related to such results (i.e., temporal modeling performance).  Since most baselines did not release their code publicly, we can only compare several models.
>
> For the question about labels for optimizing progression prediction, we adopt labels provided by Chest ImaGenome, which is also extracted from the MIMIC-CXR dataset and can be assessed at https://physionet.org/content/chest-imagenome/1.0.0/

---

### Meta-Review · Area_Chair_uw2o · 2023-09-19

**Recommendation:** 4

**Metareview:**

The paper introduces a method for generating radiology reports by incorporating both the current radiology image and the patient's image from a previous visit. The proposed technique employs a two-stage training approach: initially, the model predicts disease observations and progressions, which are utilized to construct a disease progression graph. Subsequently, a decoder is trained in the second stage to formulate the report based on the data obtained from the first stage.

Reviewers largely concur that the structural modeling of spatial and temporal progression aspects in radiology images is both innovative and intuitive. Robust experiments demonstrate a marked improvement over existing baselines, and in-depth ablation studies have been conducted. Nonetheless, there are reservations concerning the complexity of the proposed method and its potential adaptability to datasets other than MIMIC-CXR.

---

### Decision · Program_Chairs · 2023-10-07

**Decision:**

Accept-Findings

**Comment:**

The paper introduces a method for generating radiology reports by incorporating both the current radiology image and the patient's image from a previous visit. The proposed technique employs a two-stage training approach: initially, the model predicts disease observations and progressions, which are utilized to construct a disease progression graph. Subsequently, a decoder is trained in the second stage to formulate the report based on the data obtained from the first stage.

Reviewers largely concur that the structural modeling of spatial and temporal progression aspects in radiology images is both innovative and intuitive. Robust experiments demonstrate a marked improvement over existing baselines, and in-depth ablation studies have been conducted. Nonetheless, there are reservations concerning the complexity of the proposed method and its potential adaptability to datasets other than MIMIC-CXR.